# Brain Effects of *SC*-Nanophytosomes on a Rotenone-Induced Rat Model of Parkinson’s Disease—A Proof of Concept for a Mitochondria-Targeted Therapy

**DOI:** 10.3390/ijms232012699

**Published:** 2022-10-21

**Authors:** Daniela Mendes, Francisco Peixoto, Maria Manuel Oliveira, Paula Branquinho Andrade, Romeu António Videira

**Affiliations:** 1REQUIMTE/LAQV, Laboratory of Pharmacognosy, Department of Chemistry, Faculty of Pharmacy, University of Porto, Rua de Jorge Viterbo Ferreira, nº 228, 4050-313 Porto, Portugal; 2Chemistry Center-Vila Real (CQ-VR), Biological and Environment Department, School of Life and Environmental Sciences, University of Trás-os-Montes e Alto Douro, UTAD, P.O. Box 1013, 5001-801 Vila Real, Portugal; 3Chemistry Center-Vila Real (CQ-VR), Chemistry Department, School of Life and Environmental Sciences, University of Trás-os-Montes e Alto Douro, UTAD, P.O. Box 1013, 5001-801 Vila Real, Portugal

**Keywords:** anthocyanins, algae lipids, mitochondria, neurodegenerative diseases, nanomedicine

## Abstract

Mitochondria are an attractive target to fight neurodegenerative diseases due to their important functions for cells and the particularly close relationship between the functional connectivity among brain regions and mitochondrial performance. This work presents a mitochondria-targeted therapy designed to modulate the functionality of the mitochondrial respiratory chain and lipidome, parameters that are affected in neurodegeneration, including in Parkinson’s disease (PD). This therapy is supported by *SC*-Nanophytosomes constructed with membrane polar lipids, from *Codium tomentosum*, and elderberry anthocyanin-enriched extract, from *Sambucus nigra* L. *SC*-Nanophytosomes are nanosized vesicles with a high negative surface charge that preserve their properties, including anthocyanins in the flavylium cation form, under conditions that mimic the gastrointestinal tract pH changes. *SC*-Nanophytosomes, 3 µM in phospholipid, and 2.5 mg/L of EAE-extract, delivered by drinking water to a rotenone-induced PD rat model, showed significant positive outcomes on disabling motor symptoms associated with the disease. Ex vivo assays were performed with two brain portions, one comprising the basal ganglia and cerebellum (BG-Cereb) and the other with the cerebral cortex (C-Cortex) regions. Results showed that rotenone-induced neurodegeneration increases the α-synuclein levels in the BG-Cereb portion and compromises mitochondrial respiratory chain functionality in both brain portions, well-evidenced by a 50% decrease in the respiratory control rate and up to 40% in complex I activity. Rotenone-induced PD phenotype is also associated with changes in superoxide dismutase and catalase activities that are dependent on the brain portion. Treatment with *SC*-Nanophytosomes reverted the α-synuclein levels and antioxidant enzymes activity to the values detected in control animals. Moreover, it mitigated mitochondrial dysfunction, with positive outcomes on the respiratory control rate, the activity of individual respiratory complexes, and the fatty acid profile of the membrane phospholipids. Therefore, *SC*-Nanophytosomes are a promising tool to support mitochondria-targeted therapy for neurodegenerative diseases.

## 1. Introduction

The development of effective therapies to fight neurodegeneration is a major scientific challenge and an imperative need to ensure healthy lives and promote well-being for all ages. In fact, at least 50 million people are suffering from a neurodegenerative pathology, with Alzheimer’s (AD) and Parkinson’s (PD) being the most common diseases, without any reliable treatment to stop or slow the progression of the degenerative process [1,2]. 

Despite the fog that still surrounds the molecular mechanisms that trigger the neurodegenerative process, it is known that a successful therapy should be able to restore the functional connectivity among brain regions lost during the neurodegenerative process as a consequence of distorted and/or disrupted neural networks with progressive severity. The functional connectivity of brain regions is highly dependent on mitochondrial performance, as indicated by the strong spatiotemporal correlation between the synchronized neuronal activity in the γ frequency band (electrocorticogram, ECoG connectivity) and the high oxygen and glucose consumption patterns (blood-oxygen-level-dependent, BOLD connectivity) [3]. It is well-known that mitochondrial dysfunction, connected with abnormal respiratory chain activity, is a cellular hallmark of many neurodegenerative diseases [4,5,6,7,8]. In AD and PD, mitochondrial dysfunction is mainly detected by a deficient complex I (NADH:ubiquinone oxidoreductase) activity and by an abnormal membrane lipid profile [9,10]. Additionally, a dysfunctional mitochondrial redox chain affects the bioenergetic performance of the cell and the ATP-dependent metabolic pathways, increasing the generation of reactive oxygen species (ROS) that trigger profound changes in the cellular signaling cascades, promoting oxidative stress, chronic inflammation, and the accumulation of misfolded proteins—other pathological marks of diseases [11,12,13,14]. Mitochondrial dysfunction may have a negative impact on cell-to-cell communication processes, compromising brain connectivity before promoting the death of individual cells. In fact, there is scientific evidence for many neurodegenerative diseases that a massive synapse loss precedes the neurons’ death [9]. Thus, the recognition of the importance of mitochondria in brain intercellular networks is shifting the paradigm of neurodegeneration therapies toward mitochondria. 

Regarding PD and AD, the first generation of mitochondria-targeted therapy, supported by antioxidant molecules, was designed to combat the oxidative damage resulting from the overproduction of ROS by a dysfunctional mitochondrial redox chain. Despite their positive outcomes in animal models, the results obtained in clinical trials were disappointing (for review, see [15]).

We are addressing the mitochondria-targeted therapy challenge from a different perspective, rationalized on the structural and functional mitochondrial parameters that are affected by neurodegeneration [15,16], that in PD and AD are the impaired complex I activity and the changes in mitochondrial lipidome. Thus, a lipid-based nanoplatform, named *SC*-Nanophytosomes, was built with *Codium tomentosum* membrane polar lipids and elderberry anthocyanins-enriched extract (EAE-extract) from *Sambucus nigra* [16]. Elderberry anthocyanins were selected by their ability to oxidize NADH in the aqueous phase and deliver the electrons to complex III of the mitochondrial redox chain, bypassing the abnormalities associated with impaired complex I activity working as a membrane electron carrier [17]. *C. tomentosum* membrane polar lipids, with high levels of anionic phospholipids (including 22.5% of phosphatidylglycerol, which is a precursor of cardiolipins) and omega-3 polyunsaturated fatty acids (PUFA) [16,18], were selected to modulate the mitochondrial lipidome. Additionally, the high levels of anionic phospholipids with PUFA, specific from chloroplast-rich tissues of green algae and plants, are required to preserve in the bilayer of the nanovesicles the elderberry anthocyanins in their mitochondriotropic cationic form [16]. In vitro assays with neuronal cells revealed that *SC*-Nanophytosomes are competent to target mitochondria and to improve the activity of the mitochondrial redox chain, protecting the cells against the toxicity induced by rotenone or glutamate—toxins with relevance in the context of neurodegeneration [16]. In this way, our previous results allow us to move forward to investigate the ability of *SC*-Nanophytosomes to work as a mitochondria-targeted therapy in vivo when delivered by oral route. A rotenone-induced rat model of PD that mimic many behavioral and neuropathological features of the human disease, including mitochondrial complex I impairment [19,20], was used to obtain a proof-of-concept.

## 2. Results and Discussion

### 2.1. SC-Nanophytosomes Formulation Stability under pH Changes That Mimic the Gastrointestinal Tract Environments

The development of lipid-based nanoformulation to target the brain by oral administration requires nanoplatforms with the competence to overcome several physiological barriers. The nanoformulation should preserve its structural and functional properties throughout the digestive tract to maximize its uptake at different compartments (e.g., buccal cavity, small intestine). From the mouth to the small intestine, the *SC*-Nanophytosomes are subjected to different external conditions, including pH changes from 6.2–7.4 values (pH of saliva) to the high acidic conditions in the stomach (pH 1.0–2.0) and then for quasi-neutral pH environment along the small intestine (pH 6.0–7.5) [21,22]. Thus, to evaluate the impact of the pH changes throughout the gastrointestinal tract on the *SC*-Nanophytosomes stability, the formulation was submitted to successive dialysis against an aqueous buffer solution with pH values representative of the above-mentioned gastrointestinal tract compartments. The effects on *SC*-Nanophytosomes properties were assessed regarding the entrapment efficiency (EE) of the EAE-extract, vesicle size, and surface charge (Figure 1).

*SC*-Nanophytosomes, prepared with a concentration of 600 µM of algae membrane polar lipids and 0.5 mg/mL of EAE-extract, exhibit the typical absorption bands of elderberry phenolics with a maximum at 280 (as the shoulder), 350, and 520 nm that are preserved during the successive dialysis against buffers with pH of 6.4, 2.0 and 7.4 (Figure 1B). The absorption band with a maximum at 520 nm is typical of the anthocyanins in flavylium cation form (acidic form) since its blue quinoidal bases, the dominant form in aqueous solution with pH > 3, exhibited a maximum at 430 nm [23]. Therefore, the pH changes in the external environment do not affect anthocyanins in the *SC*-Nanophytosomes, suggesting that they form a stable complex with algae membrane polar lipids. Moreover, at a quantitative level, the anthocyanins are preserved in the vesicles during the three steps of dialysis, as indicated by the EE value of 81.39 ± 0.43% after the dialysis against buffer with pH 6.4, 76.01 ± 1.77% after the dialysis against buffer with pH 2.0, and 68.33 ± 4.69% after the dialysis against buffer with pH 7.4 (Figure 1A). The size and surface charge of the dialyzed *SC*-Nanophytosomes suspensions, evaluated by DLS and ELS, respectively, at pH 6.4, 2.0, and 7.4, are also displayed in Figure 1A. DLS data indicate that *SC*-Nanophytosomes dialyzed at pH 6.4 have an average diameter of 96.84 ± 14.28 nm, and that size increases to 129.94 ± 11.82 nm after dialysis at pH 2.0 and for 130.74 ± 21.22 nm by the additional dialysis step against an aqueous solution with pH 7.4. Despite the detected fluctuations in the average diameter exhibited by vesicles in response to the external pH changes, the differences did not reach statistical significance. In fact, the typical plots of the size distribution, expressed by number in terms of relative abundance, of *SC*-Nanophytosomes dialyzed against pH 6.4, 2.0, and 7.4 buffer solutions also reveal vesicle size stability (Figure 1C–E). Zeta potential measurements show that dialyzed *SC*-Nanophytosomes have a strongly negative surface charge, with a Zeta potential value of −45.66 ± 9.14 mV after dialyzed at pH 6.4, −38.69 ± 6.09 mV after dialyzed at pH 2.0 and −40.30 ± 5.53 mV after dialyzed at pH 7.4 (Figure 1A). In accordance with our previous work [16], the *SC*-Nanophytosomes exhibit a strongly negative surface charge but preserve the anthocyanins in flavylium cation form. In general, data in Figure 1 shows that *SC*-Nanophytosomes formulation is stable under the pH changes that mimic the gastrointestinal tract environments.

### 2.2. Effects of SC-Nanophytosomes on Rotenone-Induced Parkinson’s Disease Animal Model—Behavior Benefits and α-Synuclein Outcomes

To obtain a proof-of-concept for our mitochondria-targeted therapy for neurodegenerative diseases, an experimental design was rationalized considering a suitable animal model, the route of *SC*-Nanophytosomes administration, and the concentration (dose) used. A rotenone-induced PD animal model was chosen since it is highly reproducible and mimics PD-like motor deficits, including resting tremor and postural instability, as well as the progressive degeneration of the nigrostriatal dopamine system. Additionally, at a cellular level, it displays many pathological marks of PD, including impairment of mitochondrial complex I activity, accumulation of α-synuclein-containing Lewy bodies, and oxidative stress [19,20]. Thus, it exhibits a pattern of changes that reflect impairment in functional connectivity involving different brain regions and specific peripheral tissues, as well as in the bioenergetic performance of the cells that support functional networks. *SC*-Nanophytosomes formulation delivered by drinking water at 3 µM in phospholipid and 2.5 mg EAE-extract/L was chosen since the aqueous suspensions *SC*-Nanophytosomes are stable for 14 days [16]. The formulation is not affected by pH changes throughout the gastrointestinal tract, and membrane phospholipids-based vesicles at 3 µM in phospholipid are considered highly compatible with the physiological conditions [24]. Additionally, *SC*-Nanophytosomes delivered by drinking water allow *SC*-Nanophytosomes uptake in different compartments of the gastrointestinal tract, including in the buccal cavity, with privileged access to the brain [25].

*Wistar albino* rats injected with rotenone (ROT group, n = 10) three times a week for three weeks developed a PD-like phenotype with postural instability and bradykinesia (Appendix A), that compromised their performance in beam walking test, as detected by an increased in the time required to complete the task (t = 7.25 ± 2.22 s) when compared with control animals (CTRL group, n = 5, t = 3.13 ± 0.33 s) (Figure 2B). Then, animals with PD-like phenotype were randomly divided into two groups with five animals per group, and the experimental plan followed with CTRL, ROT, and ROT + *SC*-Nanophyt groups. 

During the three weeks of treatment with *SC*-Nanophytosomes, the time required by the animals of the three groups to complete the beam walking test was measured once a week (Figure 2C–E). Data show that the animals treated with *SC*-Nanophytosomes (ROT + *SC*-Nanophyt group) exhibit better performance in the beam walking test compared with animals of the ROT group, and the differences concerning the CTRL group are also attenuated. Moreover, the *SC*-Nanophytosomes benefits are detected at the final of the first week and sustained during the three weeks of treatment. It is essential to highlight that two weeks after the induction of PD-like pathology, the ROT group showed an improvement in test performance (Figure 2B,C), suggesting that part of the motor deficits detected in the first measurement can be attributed to acute toxicity of rotenone. In fact, one week of a break between the last rotenone injection and the beginning of the *SC*-Nanophytosomes treatment was rationalized to allow the rotenone clearance from the organism and to test the effects on the rotenone-induced neurodegenerative process. The obtained results indicated that the rotenone-induced PD animal model exhibits significant motor deficit, at least for four weeks, and that our *SC*-Nanophytosomes formulation, delivered by oral route, promotes significant benefits on motor coordination and balance disturbance. It is also important to highlight that, after three weeks of treatment, the animal’s body weight of the ROT + *SC*-Nanophyt group, as well as the weight of their peripheral organs (e.g., heart, kidneys, and liver), did not show significant differences compared with the CTRL group (Appendix A). Additionally, these animals no longer display visual signs of postural instability and bradykinesia or of any other health problems (Appendix A).

Regarding the classical PD model, the dopaminergic degeneration at substantia nigra pars compacta triggers the disease. However, experimental evidence supports the idea that the cerebellum is also involved in PD pathology, while reciprocal connections between the basal ganglia and cerebellum were revealed by anatomical studies [26]. Additionally, the degeneration of dopaminergic nigrostriatal pathways in PD also disrupts the connectivity between the motor cortex and substantia nigra pars compacta [27,28]. Thus, the biochemical studies described below were performed in two brain portions with similar weight (Appendix A): one comprising the basal ganglia and cerebellum (BG-Cereb) and the other with the cerebral cortex (C-Cortex) regions.

α-synuclein is a cytosolic protein expressed in brain cells, mainly in the hippocampus, substantia nigra, thalamus, cerebellum, and neocortex. In physiological conditions, it occurs in soluble monomeric form playing multiple functions, including synaptic vesicle trafficking and regulation of dopamine metabolism. However, the abnormal intracellular aggregation of misfolded α-synuclein leads to the formation of intracellular inclusions named Lewy bodies, which is a pathological hallmark in PD [6,29]. Thus, the levels of α-synuclein in rat brain BG-Cereb and C-Cortex portions were evaluated, and the results are displayed in Figure 3A,B. Data show that the brain levels of α-synuclein in the BG-Cereb portion are significantly higher than in the C-Cortex portion. Moreover, only the BG-Cereb brain portion of the ROT group exhibits a higher level of α-synuclein in relation to the CTRL group. Therefore, the enhancement of this pathological mark in the rotenone-induced PD animal model seems to be selective for the brain regions where the α-synuclein is highly expressed under physiological conditions. The *SC*-Nanophytosomes treatment (ROT + *SC*-Nanophyt group) promoted a significant decrease in α-synuclein levels in the BG-Cereb, reestablishing the levels of the CTRL group (Figure 3A). These results support the use of the rotenone toxic stimuli to induce a PD-like phenotype in rats and highlight the beneficial effect of the *SC*-Nanophytosomes treatment on this brain pathological mark of PD.

Tyrosine hydroxylase is a rate-limiting enzyme in dopamine biosynthesis, and a decrease in the tyrosine hydroxylase-positive neurons is considered a marker of nigrostriatal dopamine system degeneration. Thus, the activity of the tyrosine hydroxylase in rat brain BG-Cereb and C-Cortex portions was evaluated, and the results are displayed in Figure 3C,D. In both brain portions, tyrosine hydroxylase activity did not show significant differences among the three groups. The absence of changes in the brain tyrosine hydroxylase activity between CTRL and ROT groups suggests that our rotenone-induced PD rat model (3.0 mg of rotenone/kg of animal weight, three times a week for three weeks) does not exhibit substantial loss of dopaminergic neurons, despite their deterioration in small brain regions cannot be excluded. In fact, the loss of tyrosine hydroxylase-positive neurons in substantia nigra resulting from rat treatment with rotenone (2.75–3.0 mg/kg/day, up to 60 days) was previously reported [30]. However, it was also reported that the treatment of rodents with rotenone (2.0–4.0 mg/kg/day, for 45 days) does not promote significant changes in the density of the tyrosine hydroxylase-positive neurons, despite the well-evidenced debilitating PD-phenotype exhibited by animals [31]. Therefore, the rotenone-induced PD-like phenotype can be more consistently related to changes in brain connectivity and motor dysfunction than to the loss of dopaminergic neurons in the substantia nigra.

### 2.3. Effects on Brain Mitochondria

Rotenone-induced PD animal model exhibits impaired mitochondrial respiratory function, detected by deficient oxidative phosphorylation and by decreased complex I activity [32], which are also pathological hallmarks of PD [33]. Thus, the competence of the oral administration of *SC*-Nanophytosomes to work as a brain mitochondria-targeted therapy was assessed considering effects: (i) on the mitochondrial respiratory function under conditions that ensure the coupling between mitochondrial respiratory chain and ATP synthesis (Figure 4), (ii) on the activity of individual respiratory complexes normalized by the activity of citrate synthase (CS) (Figure 5), and (iii) on the fatty acid profile of mitochondrial membrane polar lipids (Figure 6 and Figure 7).

Figure 4 shows the results of the brain mitochondrial respiration parameters of the BG-Cereb portion for the three animal groups obtained with fresh preparations. To simulate the physiological conditions, mitochondrial respiration was supported by pyruvate/malate plus succinate, and ADP was used to induce state 3, characterized by a high oxygen consumption rate associated with ATP synthesis. After total ADP phosphorylation, the oxygen consumption rate decreased, reaching a new plateau used to assess respiration in state 4. The oxygen consumption not connected with mitochondrial respiration was assessed after adding sodium azide to inhibit complex IV (cytochrome c oxidase). The RCR, used as a measurement of the mitochondrial phosphorylation efficiency, is calculated as the ratio between the oxygen consumption rate in states 3 and 4. Although the oxygen consumption rates in states 3 and 4 do not exhibit significant differences among the three animal groups, the mitochondria of the ROT group exhibit a significant decrease in RCR compared with the CTRL group. On the other hand, the RCR value of ROT + *SC*-Nanophyt group mitochondria is comparable to those found for the CTRL group and significantly higher than that of ROT group mitochondria. Therefore, the phosphorylation efficiency of the brain mitochondria of the rotenone-induced PD animal model is compromised, and *SC*-Nanophytosomes treatment revealed competence to recover it for the values of control animals.

Mitochondrial electron transport chain complexes I, II (succinate dehydrogenase), and IV activities were evaluated in the mitochondria-rich fraction obtained from both BG-Cereb and C-Cortex brain portions (Figure 5). The activities of these complexes were normalized by the activity of CS, expressed by nmol citrate per min per mg of protein, to discard the influence of putative differences in the mitochondria content of sample preparations. In both brain portions, the values of CS activity do not have significant differences among the tested groups (BG-Cereb: CS values of 368.39 ± 43.89, 323.31 ± 40.57, 413.74 ± 112.77 for CTRL, ROT and ROT + *SC*-Nanophyt groups, respectively; C-Cortex: CS values of 444.14 ± 99.03, 452.57 ± 63.50, 469.23 ± 65.60 for CTRL, ROT, and ROT + *SC*-Nanophyt groups, respectively). Considering that the CS activity values are tightly correlated with morphometric data [34], we can also conclude that the mitochondrial content is similar in both brain portions.

A general view of the results of Figure 5 indicates that, in both brain portions, mitochondrial complex I, II, and IV activities are decreased in the ROT group compared to the CTRL group and that rats’ treatment with *SC*-Nanophytosomes recovered the activities of these respiratory complexes. Additionally, the impaired activity of the complex I associated with the deterioration of other complexes’ activity, detected in both brain portions of the ROT group, suggests that the rotenone promotes a neurodegenerative process with a mitochondrial dysfunction extendable to the entire brain that goes beyond the expectable inhibitory effects on complex I. Despite this, the brain effects of mitochondrial dysfunction promoted by rotenone and *SC*-Nanophytosomes treatment can also be dependent on the brain region, since, under physiological conditions (CTRL group), the BG-Cereb mitochondria exhibit a respiratory redox chain with a 40% greater capacity to process the NADH produced in the citric acid cycle than mitochondria of C-Cortex portion. In fact, a detailed analysis of Figure 5 shows that the results depend on the mitochondrial complex and the brain portion. Regarding complex I activity (Figure 5A,D), the ROT group has a significant decrease in both brain portions, suggesting that the impairment of complex I promoted by rotenone is not restricted to any particular brain region. On the other hand, the *SC*-Nanophytosomes treatment recovered the complex I activity when compared to the CTRL group. However, the differences between ROT + *SC*-Nanophytosomes and ROT groups only reached statistical significance in the BG-Cereb brain portion. Complex II has lower activity in the ROT group, but the significant differences are only detected in comparison to the CTRL group in the BG-Cereb portion (Figure 5B,E). On the other hand, the complex IV activity in the ROT group is significantly lower than in the CTRL group in the C-Cortex (Figure 5C,F). In this brain portion, the complex IV activity in the ROT + *SC*-Nanophyt group is also significantly higher than in the ROT group. These results indicate that *SC*-Nanophytosomes formulation has the competence to restore/attenuate the impairment of mitochondrial respiratory complexes activity exhibited by the rotenone-induced PD animal model. Using in vitro models, we have shown that elderberry anthocyanins are redox-active mitochondriotropic compounds with the ability to overcome the complex I inhibition promote by rotenone [17] and that *SC*-Nanophytosomes use the caveola-mediated endocytosis pathway to target mitochondrial, protecting the neuronal cells against the glutamate- and rotenone-induced toxicity [16]. Therefore, the present in vivo results indicates that *SC*-Nanophytosomes, delivered by drinking water at 3 µM in phospholipid and 2.5 mg EAE-extract/L for three weeks, ensure the targeting of the elderberry anthocyanins to the mitochondria of brain cells, allowing the redox-active anthocyanins to reach brain concentrations with therapeutic efficacy.

Considering that *SC*-Nanophytosomes were built with elderberry anthocyanins and *C. tomentosum* membrane polar lipids, the next step was to investigate if the algae lipids, in addition to the nanoplatform physical support, also have therapeutic benefits by modulating the mitochondrial lipidome. Thus, the fatty acid profile of the brain mitochondria phospholipids was revealed for three animal groups (CTRL, ROT, and ROT + *SC*-Nanophyt), with results discriminated for both BG-Cereb and C-Cortex brain portions as displayed in Figure 6 and Figure 7. *C. tomentosum* membrane polar lipids are characterized by a fatty acid profile with high levels of PUFA (34%) and an n-3/n-6 PUFA ratio of 1.5 ± 0.3 [16]. Therefore, an enrichment of the mitochondrial lipidome in PUFA leads to an increase in unsaturation index and to the enhancement of the n-3/n-6 PUFA ratio, which can be used as indicators of the *SC*-Nanophytosomes’ ability to deliver algae lipids to brain mitochondria. In fact, the functional organization of the mitochondrial redox chain complexes requires phospholipid bilayers with fatty acids with high unsaturation index, mainly ensured by mature cardiolipin species with three/four PUFA [9,35]. The n-3/n-6 PUFA and DHA/AA ratios balance are important players in the many physiological processes, and the changes in those ratios are associated with neuroinflammation and neurodegenerative diseases, including PD [36,37,38]. Additionally, it was also described that diets enriched in n-3 PUFAs and/or in DHA/AA ratios improve cognitive functions and exert neuroprotective effects in PD animal models [39,40,41,42].

Data from Figure 6 show that mitochondria from both brain portions of the CTRL group exhibit a fatty acid profile with twenty-eight fatty acids species, comprising ten saturated fatty acids (SFA), six monounsaturated fatty acids (MUFA), and twelve PUFA. Regarding SFA, palmitic acid (C16:0) is the most abundant SFA followed by stearic acid (C18:0), while oleic acid (C18:1) is the dominant specie of PUFA. Regarding PUFA, docosahexaenoic acid (DHA, C22:6n-3) is the most abundant molecular species in the n-3 series, while arachidonic acid (AA, C20:4n-6) is the most abundant in the series n-6. Eicosadienoic (C20:2), linoleic (C18:2n-6) and docosatetraenoic (C22:4n-6) acids also have relative abundances higher than 1%. The rotenone-induced PD-like pathology promotes minor changes in the mitochondrial fatty acid profile. Compared to CTRL, the ROT group presents a significant increase in palmitoleic acid (C16:1) content in the BG-Cereb portion. In contrast, in the C-Cortex portion, the major differences occur at the level of C16:0 (increase) and C22:6n-3 (decrease). The treatment with *SC*-Nanophytosomes (ROT + *SC*-Nanophyt group) reverts the rotenone effects by decreasing C16:1 and C16:0 levels in BG-Cereb and C-Cortex portions, respectively. Additionally, the relative abundance of C22:6n-3 in the C-Cortex portion is increased, surpassing the CTRL level. The quantitative data in Figure 6 were used to determine the total levels of SFA, MUFA, n-3, and n-6 PUFA, as well as to calculate the unsaturation index, and the n-3/n-6 PUFA and DHA/AA ratios. The obtained results are displayed in Figure 7. In the BG-Cereb portion, the small changes in the levels of individual fatty acid species detected among the three experimental groups did not lead to significant changes in the unsaturation index nor in the n-3/n-6 PUFA and DHA/AA ratios. However, in the C-Cortex portion, the mitochondria of the ROT group exhibit a fatty acid profile with a lower unsaturation index and with lower n-3/n-6 PUFA and DHA/AA ratios, as compared with the CTRL group. Thus, the functional heterogeneity of the different brain portions is evidenced, associated with different lipid profiles and responses according to the function. The PD phenotype presented notorious lipid changes mainly on the C-Cortex fraction. When treated with *SC*-Nanophytosomes (ROT + *SC*-Nanophyt group), the fatty acid profile was reverted to CTRL levels or even improved, as observed for the unsaturation index, n-3/n-6 PUFA, and DHA/AA ratios (Figure 7). These results indicate that the *SC*-Nanophytosomes are competent to modulate the lipidome of the brain mitochondria, promoting positive outcomes in rotenone-induced PD-like pathology.

### 2.4. Effects on the Brain Redox State 

The brain redox state was evaluated in mitochondria-free cytosolic fraction obtained from rat BG-Cereb and C-Cortex brain portions regarding the levels of reduced glutathione (GSH) and oxidized glutathione (GSSG) (Figure 8) and the activity of antioxidant enzymes, namely superoxide dismutase (SOD), catalase (CAT), glutathione reductase (GR), and glutathione peroxidase (GPx) (Figure 9). The concerted activity of these antioxidant enzymes promotes the management of the ROS simultaneous with the dynamic regulation of the GSH/GSSG ratio, used as an indicator of cell redox state [43,44,45]. Therefore, the results of Figure 8 and Figure 9 are discussed in an integrated fashion to provide an overview of the tissue’s ability to respond to oxidative stress stimuli. 

Data in Figure 8 indicate that the cells of the C-Cortex have a cytosolic environment significantly more reduced (a redox potential more negative) than the cells of the BG-Cereb portion since the values of the GSH/GSSG ratio in the C-Cortex portion are about three times higher than in BG-Cereb. Additionally, the total cytosolic content of glutathione (GSH + 2GSSG) is also significatively greater in C-Cortex cells, suggesting a high ability to handle ROS. In both brain portions, no significant differences were detected for GSH/GSSG ratio between ROT and CTRL animal groups (Figure 8C,F), suggesting that four weeks after the development of rotenone-induced PD-like phenotype (ROT group) the oxidative damage is not yet a pathological hallmark spread throughout the brain. Consequently, treatment with *SC*-Nanophytosomes of rats with a PD-like phenotype (ROT + *SC*-Nanophyt group) did not promote detectable effects on the cytosolic redox state of brain cells. Despite this, the comparison of the activity of the antioxidant enzymes in three animal groups (Figure 9) suggests that the brains of the animals with rotenone-induced PD-like pathology regulate the activity of these enzymes in response to oxidative stress stimuli. 

In the BG-Cereb brain portion (Figure 9A–D), SOD, CAT, and GR activity decreased in the ROT group compared to the CTRL group (significant differences were detected for CAT and GR). While for GPx activity, no significant differences were detected among the three groups, the values of the activity of SOD, CAT, and GR enzymes in the ROT + *SC*-Nanophyt group are not significantly different from those detected in the CTRL group. However, they are higher than those observed in the ROT group, with statistical significance for SOD and GR. Therefore, the rise of the vulnerability to oxidative stress exhibited by the ROT group is attenuated by treatment with *SC*-Nanophytosomes.

In the C-Cortex brain portion (Figure 9E–H), the effects of the rotenone-induced PD-like pathology on the activity of SOD and CAT follow a pattern with an opposite behavior to the observed in the BG-Cereb brain portion. Thus, the ROT group exhibits significantly higher activity of SOD and CAT enzymes than the CTRL group. Consistently, in the ROT + *SC*-Nanophyt group, the values of the activity of SOD and CAT are not significantly different from those detected in the CTRL group. However, they are lower than those observed in the ROT group despite the differences did not reach statistical significance. The activity of GR and GPx enzymes did not show significant differences among the three groups.

From the data in Figure 8 and Figure 9, we can conclude that the treatment with *SC*-Nanophytosomes has the competence to attenuate and/or counteract the effects of the rotenone-induced PD-like pathology on the antioxidant enzymes of the cytosolic fraction of cells obtained from both brain portions. Additionally, the rotenone-induced PD rats have a BG-Cereb brain portion more vulnerable to oxidative damage than the C-Cortex brain portion.

## 3. Material and Methods

### 3.1. SC-Nanophytosomes Building Blocks—Codium Tomentosum Polar Membrane Lipids and Elderberry Anthocyanins-Enriched Extract

*Codium tomentosum* polar membrane lipids were obtained from seaweed harvested in October 2019 at the beach of Aguda, Vila Nova de Gaia (41°02′58.9″ N, 8°39′20.4″ W), by using our green methodology described elsewhere [16]. Elderberry anthocyanins-enriched extract (EAE-extract) was obtained from fresh mature fruits (ca. 1 kg), harvested in September 2019 in northern Portugal (40°46′34″ N, 7°3′46″ W) from five plants of European elderberry (*Sambucus nigra*), according with a methodology previously described [17]. The detailed characterization of both building blocks was disclosed in our previous report [16].

### 3.2. SC-Nanophytosomes Formulation Preparation and Characterization under Conditions That Mimic the Gastrointestinal Tract

*SC*-Nanophytosomes were prepared with EAE-extract (0.5 mg/mL) and *C. tomentosum* membrane polar lipids (600 μM in phospholipids), using a procedure to preserve anthocyanins in their flavylium cation form [16]. The oral route will be used to evaluate the therapeutic competence of *SC*-Nanophytosomes formulation in a rotenone-induced rat model of PD. Thus, the effects of pH changes on the properties of *SC*-Nanophytosomes were evaluated by a successive dialysis process through a cellulose membrane (cut-off of 14 kDa Sigma-Aldrich, St. Louis, MO, USA) against buffer solutions that mimic the pH conditions throughout the gastrointestinal tract. The *SC*-Nanophytosomes formulation (5 mL), prepared in aqueous buffer solution (50 mM KCl, 10 mM 4-(2-hydroxyethyl)-1-piperazineethanesulfonic acid (HEPES), 2 mM citric acid, pH 6.4), was, in a first step, dialyzed against 100 mL of aqueous buffer solution (50 mM NaCl, 10 mM HEPES, 2 mM citric acid, pH 6.4) for 1h. Then, an aliquot was collected for analysis, and the remaining volume was submitted to dialysis against 100 mL of aqueous buffer solution at pH 2.0 for 1 h. An aliquot was collected, and the dialyzed formulation was submitted to new dialysis against 100 mL of aqueous buffer solution at pH 7.4 for 1 h. Aliquots of *SC*-Nanophytosomes formulation collected in the different dialysis steps were used to determine the vesicle size, surface charge, and EE of the anthocyanins. The values of the absorption at 520 nm (typical of anthocyanins in cationic form) obtained by UV-Visible spectra of non-dialyzed and dialyzed *SC*-Nanophytosomes, were used to determine the EE of elderberry anthocyanins, according to the equation: EE (%) = (Abs_dialyzed *SC*-Nanophytosomes_/Abs_non-dialyzed *SC*-Nanophytosomes_) × 100 [16]. The size and size distribution of the dialyzed *SC*-Nanophytosomes were evaluated by dynamic light scattering (DLS) using a helium-neon laser wavelength of 658 nm and a detector angle of 90° (DLS, Brookhaven Instruments, New York, NY, USA). The surface charge, assessed in terms of Zeta potential, was evaluated by electrophoretic light scattering (ELS) using the ZetaPALS instrument (Brookhaven Instruments, New York, NY, USA), as previously described [46].

### 3.3. Rotenone-Induced Rat Model of Parkinson’s Disease and Treatment with SC-Nanophytosomes

Animal experiments were conducted according to Portuguese (Decreto-Lei 113/2013) and European guidelines for Animal Care (EU Directives 2010/63/EU) and were approved by UTAD-ORBEA ethical committee. Animals were housed with food and water available *ad libitum* and maintained on a 12 h light/dark cycles in a temperature-controlled (22 °C ± 2 °C) room in the animal facilities of the University of Trás-os-Montes and Alto Douro. All efforts were made to minimize the number of animals used and their suffering. Fifteen *Wistar* rats (male) with 3-month-old were randomly assigned into two different experimental groups, one group (control) with five animals and the other group with ten animals that will be used to induce a PD-like pathology with rotenone and then subdivided into two groups according to the experimental design displayed in Figure 2A. Considering the low water solubility of the rotenone, 3 mg/mL of rotenone was solubilized in a lipid-based carrier containing 15 mM of egg yolk phospholipids. Then, rats were injected intraperitoneally (i.p.) with either rotenone (3.0 mg/kg of animal weight) formulation or vehicle (water suspension of the lipid-based carrier without rotenone) three times a week for three weeks. Then, the ten rats with rotenone-induced PD-like pathology were randomly divided into two groups with five animals each, and one week later the experimental plan followed with three groups: control (CTRL), rotenone (ROT), and rotenone treated with *SC*-Nanophytosomes delivered by drinking water at a final concentration of 3 µM in phospholipids plus 2.5 mg/L in EAE-extract (ROT + *SC*-Nanophyt). *SC*-Nanophytosomes were prepared with 600 µM membrane polar lipids from *C. tomentosum* and 0.5 mg/mL of EAE-extract from *S. nigra* in the buffer as described above and diluted with commercially available water with acidic pH (total mineralization 26 mg/L, silica 10 mg/L, bicarbonate <0.3 mg/L, chloride 7.2 mg/L, nitrate 1.3 mg/L, calcium 0.4 mg/L, sodium 4.2 mg/L and magnesium 0.7 mg/L, pH 4.7). During three weeks of the treatment, animals had free access to the liquid (water for CTRL and ROT groups, and water supplemented with *SC*-Nanophytosomes for ROT + *SC*-Nanophyt group) and standard rodent chow composed of pellets of AIN-76A balanced diet (20.3% protein, 65% carbohydrates and 5% fat, http://www.researchdiets.com, accessed on 30 August 2022). Drinking liquid (water or water supplemented with *SC*-Nanophytosomes) was changed every 3 days. Animal body weight was monitored weekly throughout the study (Appendix A).

Considering the average animals’ body weight (375.12 ± 20.69 g) and the daily liquid intake (41.10 ± 3.18 mL) for the ROT + *SC*-Nanophyt group, we have estimated an *SC*-Nanophytosomes oral dose of 300 nmol of phospholipid and 0.25 mg of EAE-extract/kg/day, which is equivalent to a human dose of 56.20 nmol of phospholipid and 46.80 mg of EAE-extract/kg/day using Food and Drug Administration criteria for converting drug equivalent dosages across species [47]. Note that humans’ recommended total polyphenol intake is about 20 mg/kg/day.

### 3.4. Beam Walking Test

The beam walking test assesses animals’ motor coordination and balance disturbance. In this test, the animal must walk across a high beam with a narrow diameter that challenges its ability to maintain balance [48]. The beam apparatus consists of a 100 cm beam with a flat surface of 6 mm (width) suspended 50 cm from the ground by wooden supports at either end. The wooden supports were put at the beginning of the beam and the end, next to which was placed a black box, serving as a home cage for the animal being tested (Figure 10). At the start of the beam, a line was drawn 20 cm that was used as starting zone. The test starts by placing the animal in the starting zone and finishes when the animal reaches the home cage. At the same time, a video camera records the performance and allows the measurement of the time that the animal takes to cross the beam (80 cm).

### 3.5. Ex Vivo Studies: Samples Collection

After three weeks of *SC*-Nanophytosomes treatment, the animals were euthanized by cervical displacement followed by decapitation. Their brains were sectioned into two different portions: one comprising the basal ganglia, a brain macro-region that includes the substantia nigra region and cerebellum (BG-Cereb), and the other with the cerebral cortex (C-Cortex) regions of the brain. The brain portions obtained were homogenized in buffer (sucrose 130 mM, KCl 50 mM, MgCl_2_ 5 mM, KH_2_PO_4_ 5 mM, and HEPES 5 mM, pH 7.4, supplemented with a protease inhibitors cocktail) using a Glass-Teflon Potter Elvejhem (20 strokes) to promote the rupture of cells. The homogenates were centrifugated at 1000× *g* for 10 min, at 4 °C, to discarded cell nuclei as a pellet. The supernatants, containing cytosolic organelles including mitochondria, were collected, and their total protein content was determined by the Biuret method [9]. One sample of each brain supernatant (1000× *g*) was collected, quickly frozen in liquid nitrogen, and stored at −80 °C until to be used for evaluation of the brain levels of α-synuclein. Another sample of the supernatant obtained with the BG-Cereb portion was immediately used to evaluate the bioenergetic performance of mitochondria with an OROBOROS Oxygraph-2k system (as described below). The remaining supernatants were fractioned by a differential centrifugation procedure to obtain mitochondria-free cytosolic and mitochondria-rich fractions containing both synaptic and non-synaptic brain mitochondria, as previously described [49]. The protein content of these sub-cellular fractions was determined by the Biuret method [9] and then kept at −80 °C until the time to be used. The heart, liver, kidneys, and samples of skeletal-muscle tissues were also collected, weighed (Appendix A), and immediately frozen in liquid nitrogen and kept at −80 °C for further studies.

### 3.6. Rat α-Synuclein (SNCA) Enzyme-Linked Immunosorbent Assay (ELISA)

α-synuclein levels were quantified in the brain supernatants in the BC-Cereb and C-Cortex brain portions by using a rat α-synuclein ELISA kit (Cusabio) according to the manufacturer’s protocol. The absorbance was read at 450 and 570 nm, and the concentrations were calculated from standard curves. Results are expressed as ng/mg of protein.

### 3.7. Tyrosine Hydroxylase Activity

Tyrosine hydroxylase activity was assessed in mitochondria-free cytosolic fraction obtained from BG-Cereb and C-Cortex portions [50]. A volume of the mitochondria-free cytosolic fraction containing 0.2 mg of protein was added to a 125 µL of HEPES buffer (100 mM, pH = 7.0) supplemented with 0.5 mM tetrahydrobiopterin and 5 µM iron (II) sulfate and the mixture incubated for 5 min. The reaction started with the addition of 125 µL of HEPES buffer supplemented with 100 µM of tyrosine (enzyme substrate) and 200 µM of sodium periodate (used to transform L-3,4-dihydroxyphenylalanine (L-DOPA) into the detectable dopachrome). The enzyme activity was followed spectrophotometrically at 475 nm, recording points every 20 s for 10 min, at 37 °C. Production of L-DOPA was determined using a molar extinction coefficient for a dopachrome of 3700 M^−1^ cm^−1^. Assays conducted in the presence of 100 μM of 3-iodo-tyrosine, a known competitive inhibitor of tyrosine hydroxylase, were used to confirm that the detected dopachrome is related to tyrosine hydroxylase activity.

### 3.8. Assessment of the Mitochondrial Respiratory Function

Brain mitochondria oxygen consumption was monitored with a Clark-type electrode, at 37 °C, in a 2 mL thermostatic chamber with continuous agitation using an OROBOROS Oxygraph-2k system (Oroboros Instruments, Innsbruck, Austria), a high-resolution respirometry apparatus that allows the detection of respiratory flux from 1 pmol.s^−1^.mL [51]. Brain supernatants (0.5–0.6 mg), containing the non-synaptic brain mitochondria in suspension and the synaptic population of brain mitochondria within synaptosomes, were added to 2 mL of reaction medium (sucrose 130 mM, KCl 50 mM, MgCl_2_ 5 mM, KH_2_PO_4_ 5 mM, and HEPES 5 mM, pH 7.4) supplemented with digitonin to a final concentration of 0.02%. Digitonin is used to permeabilize the synaptosomes membrane, allowing the assessment of oxygen consumption by both non-synaptic and synaptic brain mitochondria populations [9]. Oxygen concentration and flux were simultaneously recorded and analyzed by Dat lab software Version 7.0 (Oroboros Instruments). Fifteen minutes after starting the record, a cocktail of mitochondrial respiratory substrates (pyruvate 2.5 mM and malate 1.25 mM, and succinate 5 mM) was added to allow the activation of the mitochondrial respiratory chain at the level of complex I (pyruvate and malate are NADH generators) in simultaneous with complex II activation (succinate is the substrate of this enzyme), as occurs in the physiological conditions. To induce state 3 respiration (respiration is coupled with ATP synthesis), adenosine diphosphate (ADP 50 µM) was added. When all ADP is phosphorylated, the oxygen consumption rate decreases and the system enters state 4. Then, sodium azide 20 µM was added to inhibit complex IV, which allows the assessment of the oxygen consumption not connected with mitochondrial respiration. Thus, the mitochondrial oxygen consumption rate values in states 3 and 4, expressed by pmol O_2_/min/mg of protein, can be calculated. The respiratory control rate (RCR) is calculated as the ratio between the oxygen consumption rates in states 3 and 4. The concentrations mentioned earlier for respiratory substrates, ADP and inhibitors refer to their concentrations in the chamber. 

### 3.9. Mitochondrial Respiratory Chain Complex Activities

The mitochondria-rich fraction obtained from two brain portions was used to evaluate the activity of mitochondrial redox chain complexes I, II and IV, and CS according to our well-established procedures [49]. The activity of mitochondrial redox chain complexes was normalized by CS activity.

### 3.10. Determination of Brain Enzymatic and Nonenzymatic Antioxidant Defenses

The mitochondria-free cytosolic fraction obtained from the two brain portions was used to assess the activity of the SOD, CAT, GPx, and GR, as well as to determine the levels of GSH and GSSG, as previously described [17,49].

### 3.11. Lipid Extraction and Phospholipid Quantification

Phospholipids extraction from mitochondria-rich fraction was performed by a double extraction procedure using a combination of methanol/chloroform/water (2:1:0.8, v/v/v), as previously described [9]. Total phospholipid content was determined according to the method outlined by Bartlett and Lewis [9,52], using a standard curve ranging from 0 to 250 nmol phosphate.

### 3.12. Preparation and Analysis of Esters Derivate of Fatty Acids

Fatty acid methyl esters (FAME) from the phospholipids of each mitochondria-rich fraction extract (500 nmol in phospholipids) were obtained by acid-catalyzed transmethylation in the presence of the internal standard C17:0 (30 nmol) [53]. The obtained hexane FAME solutions were analyzed by gas chromatography using a ThermoFinningan-Trace Gas Chromatograph connected to the mass spectrometer Polaris Q MSn equipped with an Ion Trap analyzer for identification, and gas chromatography using a Finnigan Focus GC, Thermo Fisher Scientific, Waltham, MA, USA, equipped with a flame ionization detector and a VF-5 ms column (30 m×0.25 mm×0.25 μm) for quantification, as previously described [46,49]. Fatty acids were identified by comparison with retention time and fragmentation profile of reference standard mixtures FAME37 (Supelco 37 Component FAME Mix) and quantified using the peak area of the internal standard.

### 3.13. Data and Statistical Analysis

All data are presented as mean values ± standard error of the mean (SEM) of at least three independent assays. Statistical analysis was performed using GraphPad Prism 8 software (San Diego, CA, USA). The level of significance between different groups was determined by one-way analysis of variance (ANOVA), followed by the Bonferroni test (three or more sets of data) or by *t*-test (two sets of data). Statistical significance was attained at *p* < 0.05.

## 4. Conclusions

From the general overview of the results presented in the present study, it can be concluded that *SC*-Nanophytosomes have excellent properties to support mitochondria-targeted therapy for neurodegenerative diseases like PD. The characterization of *SC*-Nanophytosomes under pH conditions that mimic the gastrointestinal tract environments revealed the stability of the formulation and the ability to preserve the anthocyanins in the flavylium cation form, with mitochondriotropic properties. Thus, to obtain a proof-of-concept of the *SC*-Nanophytosomes as a mitochondria-targeted therapy, the oral administration route was rationalized to a rotenone-induced PD-like pathology animal model. The animal model used exhibits a set of phenotypic markers related to PD, comprising a significant motor deficit, higher levels of α-synuclein, compromised mitochondrial phosphorylation efficiency, impairment of the mitochondrial complex I, and an altered mitochondrial membrane lipidome, as well as changes in the cytosolic activity of the antioxidant enzymes. 

The treatment with *SC*-Nanophytosomes promoted significant benefits on motor coordination and balance disturbance, decreased α-synuclein levels to the values detected in control animals, and recovered the deficit in the mitochondrial phosphorylation efficiency and complex I activity. Furthermore, the fatty acid profile of the mitochondrial membrane phospholipids and the cytosolic activity of the antioxidant enzymes, affected by rotenone-induced PD-like neurodegeneration, were ameliorated to control levels. Combining these results with in vitro effects of elderberry anthocyanins and *SC*-Nanophytosomes [16,17], the positive outcomes on the brain mitochondrial respiratory chain can be attributed to elderberry anthocyanins, while the modulation of mitochondrial lipidome to the algae lipids. Although these results do not allow rationalize the nanoplatforms ability to cross the blood-brain barrier, they support the idea that *SC*-Nanophytosomes is an effective tool to provide elderberry anthocyanins and algae lipids for the brain cells. In this way, the present work proves the concept of *SC*-Nanophytosomes as a mitochondria-targeted therapy.

Moreover, it is expectable that *SC*-Nanophytosomes will be able to promote positive outcomes in other PD models as well as in human patients since the motor deficit, mitochondrial dysfunction and alfa-synuclein accumulation are pathological features of PD in humans, recapitulated by several chemically induced and genetic modified PD animal models (e.g., MPTP-induced, MitoPark mice) [20,54]. In this way, the present work is a good prelude to expanding the *SC*-Nanophytosomes studies aiming to unveil their safety profile, pharmacokinetic parameters in blood and brain, and therapeutic efficacy in other PD animal models. These studies are required to test in clinical trials the *SC*-Nanophytosomes as a mitochondria-targeted therapy for neurodegenerative diseases.

## Figures and Tables

**Figure 1 ijms-23-12699-f001:**
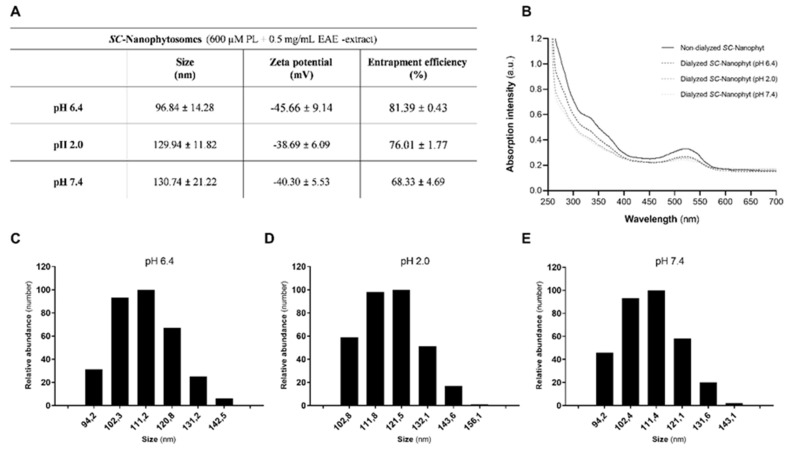
(**A**) Characterization of the dialyzed *SC*-Nanophytosomes (600 μM in phospholipid and 0.5 mg/mL EAE-extract) against buffer with different pH (6.4, 2.0, and 7.4) in terms of vesicle size (nm), Zeta potential (mV), and the entrapment efficiency (%). (**B**) Typical UV-Visible absorption spectra (250–700 nm) of the EAE-extract of non-dialyzed and dialyze *SC*-Nanophytosomes at pH 6.4, 2.0, and 7.4. (**C–E**) Typical plots of the size distribution of dialyzed *SC*-Nanophytosomes at pH 6.4 (**C**), 2.0 (**D**), and 7.4 (**E**).

**Figure 2 ijms-23-12699-f002:**
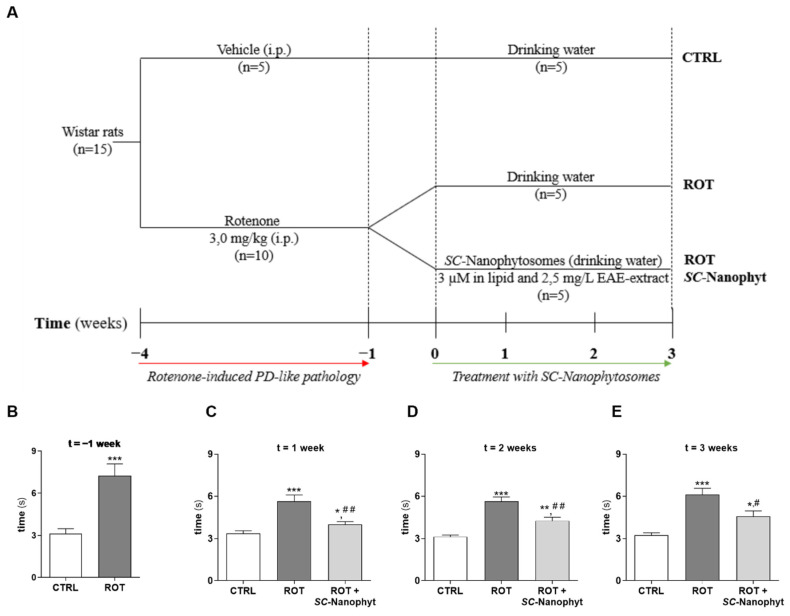
(**A**) Schematic representation of the experimental procedure. Rotenone-induced PD-like pathology for three weeks, one week break, and then three weeks of *SC*-Nanophytosomes treatment (3 µM in phospholipid and 2.5 mg/L in EAE-extract). (**B**–**E**) Average times from assessment of motor balance and coordination in the rats using the beam walking test. (**B**) Average times for rats crossing the 6 mm beam after three weeks of pre-treatment with rotenone (3.0 mg/kg, i.p.) and control. Effect of treatment with *SC*-Nanophytosomes on the first week (**C**), after two weeks (**D**), and three weeks (**E**) in average times for crossing the 6 mm beam in CTRL, ROT, and ROT + *SC*-Nanophyt groups. Error bars represent SEM for n = 5 independent experiments using five animals in each one. *, **, *** Significantly different from CTRL group, with *p* ≤ 0.05, *p* < 0.01 and *p* < 0.001, respectively. ^#^, ^##^ Significantly different from ROT group with *p* ≤ 0.05 and *p* < 0.01.

**Figure 3 ijms-23-12699-f003:**
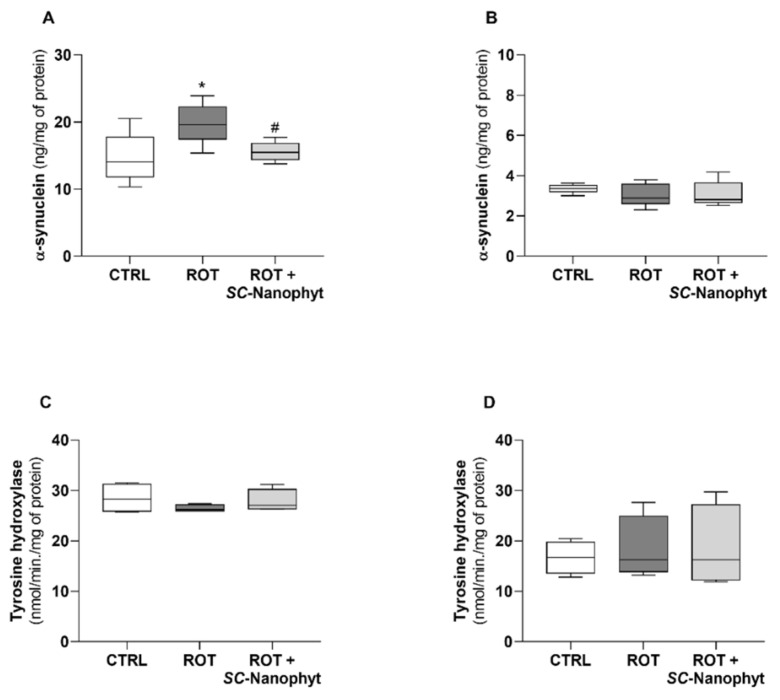
Brain α-synuclein levels assessed in brain homogenates from BG-Cereb (**A**) and the C-Cortex (**B**) portions of the CTRL, ROT, and ROT + *SC*-Nanophyt groups. The results are expressed as ng/mL/mg of protein. Tyrosine hydroxylase activity was assessed in the mitochondria-free cytosolic fraction from BG-Cereb (**C**) and the C-Cortex (**D**) portions of the CTRL, ROT, and ROT + *SC*-Nanophyt groups. The results are expressed as nmol/min/mg of protein. Error bars represent SEM for n = 5 independent experiments using five animals in each one. * Significantly different from the CTRL group, with *p* ≤ 0.05. ^#^ Significantly different from the ROT group, with *p* ≤ 0.05.

**Figure 4 ijms-23-12699-f004:**
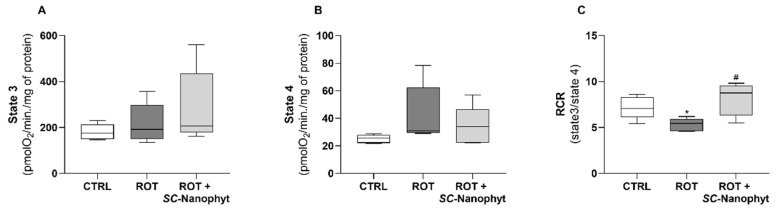
Mitochondrial respiration parameters were obtained with fresh preparations of BG-Cereb portion for CTRL, ROT, and ROT + *SC*-Nanophyt groups, supported by pyruvate/malate and succinate. The respiratory state 3 (**A**) was obtained by adding ADP to oxidize succinate, and state 4 (**B**) was reached after total phosphorylation of ADP. The RCR (**C**) was calculated as the ratio between the oxygen consumption rate in states 3 and 4. Results are expressed as pmolO_2_/min/mg of protein. Error bars represent SEM for n = 5 independent experiments using five animals in each one. * Significantly different from the CTRL group, with *p* ≤ 0.05. ^#^ Significantly different from the ROT group, with *p* ≤ 0.05.

**Figure 5 ijms-23-12699-f005:**
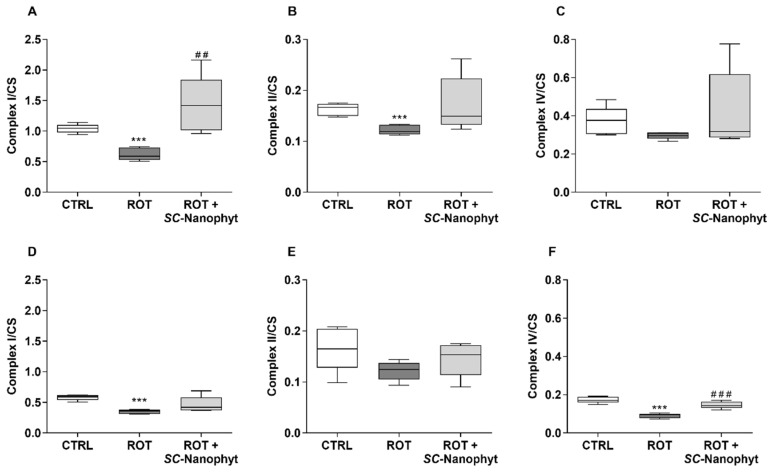
Mitochondrial complexes I (**A**,**D**), II (**B**,**E**), and IV (**C**,**F**) activity in the mitochondria-rich fraction from BG-Cereb (**A**–**C**) and C-Cortex (**D**–**F**) brain portions of CTRL, ROT, and ROT + *SC*-Nanophyt groups normalized by CS activity. Error bars represent SEM for n = 5 independent experiments using five animals in each one. *** Significantly different from CTRL group, with *p* < 0.001. ^##^, ^###^ Significantly different from ROT group, with *p* < 0.01 and *p* < 0.001, respectively.

**Figure 6 ijms-23-12699-f006:**
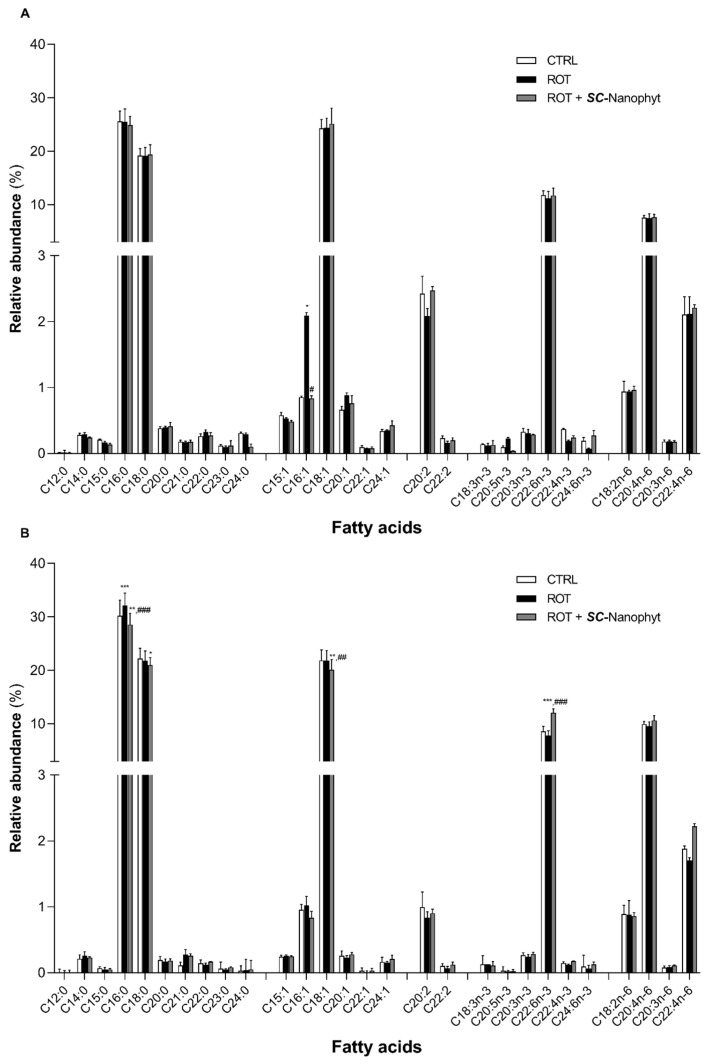
Fatty acid profile of phospholipid extracts obtained from mitochondria-rich fraction from BG-Cereb (**A**) and C-Cortex (**B**) brain portions of CTRL, ROT and ROT + *SC*-Nanophyt groups (CTRL: white bars; ROT: black bars; ROT + *SC*-Nanophyt: grey bars): C12:0 = lauric acid, C14:0 = myristic acid, C15:0 = pentadecanoic acid, C16:0 = palmitic acid, C18:0 = stearic acid, C20:0 = arachidonic acid, C21:0 = heneicosanoic acid, C22:0 = docosanoic acid, C23:0 = tricosanoic acid, C24:0 = lignoceric acid, C15:1 = pentadecenoic acid, C16:1 = palmitoleic acid, C18:1 = oleic acid, C20:1 = eicosenoic acid, C22:1 = docosenoic acid, C24:1 = nervonic acid, C20:2 = eicosadienoic acid, C22:2 = docosadienoic acid, C18:3n-3 = α-linolenic acid, C20:5n-3 = eicosapentaenoic acid (EPA), C20:3n-3 = eicosatrienoate acid, C22:6n-3 = docosahexanoic acid (DHA), C22:5n-3 = docosapentaenoic acid (DPA), C22:4n-6 = docosatetraenoate, C24:6n-3 = tetracosahexaenoic acid, C18:2n-6 = linoleic acid, C20:4n-6 = arachidonic acid (AA), C20:3n-6 = dihomo-γ-linolenic acid (DGLA). Error bars represent SEM for n = 5 independent experiments using five animals in each one. *, **, *** Significantly different from CTRL group, with *p* ≤ 0.05, *p* < 0.01 and *p* < 0.001, respectively. ^#^,^##^, ^###^ Significantly different from ROT group, with *p* ≤ 0.05, *p* < 0.01 and *p* < 0.001, respectively.

**Figure 7 ijms-23-12699-f007:**
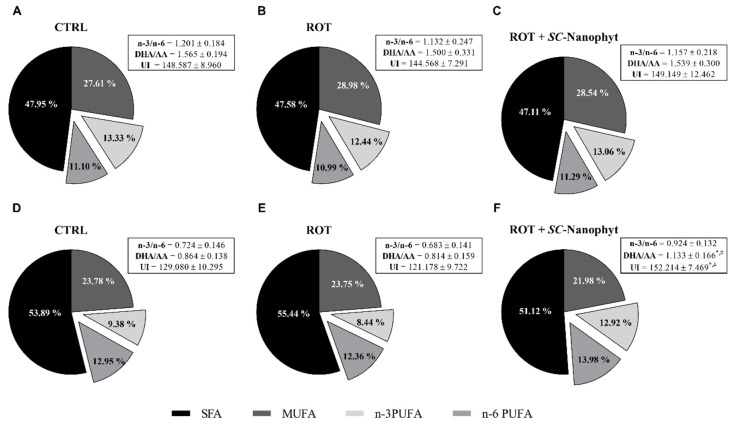
General parameters of the fatty acid profile of phospholipid extracts obtained from mitochondria-rich fraction from BG-Cereb (**A**–**C**) and C-Cortex (**D**–**F**) brain portions of CTRL, ROT and ROT + *SC*-Nanophyt groups. AA—arachidonic acid, DHA—docosahexanoic acid, SFA—saturated fatty acids, MUFA—monounsaturated fatty acids, PUFA—polyunsaturated fatty acids. * Significantly different from CTRL group, with *p* ≤ 0.05. ^#^ Significantly different from ROT group, with *p* ≤ 0.05.

**Figure 8 ijms-23-12699-f008:**
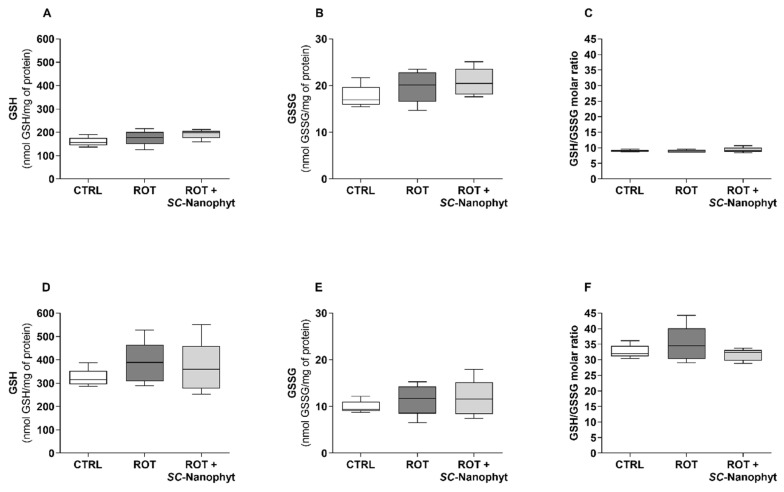
GSH (**A**,**D**) and GSSG (**B**,**E**) levels and GSH/GSSG molar ratio (**C**,**F**) in the mitochondria-free cytosolic fraction from BG-Cereb (**A**–**C**), and C-Cortex (**D**–**F**) brain portions of CTRL, ROT and ROT + *SC*-Nanophyt groups. Error bars represent SEM for n = 5 independent experiments using five animals in each one.

**Figure 9 ijms-23-12699-f009:**
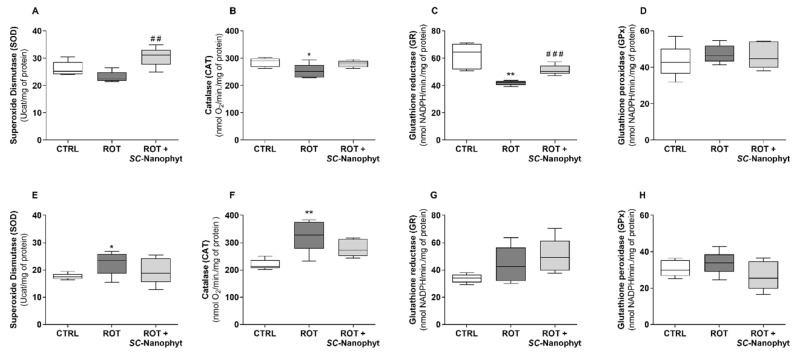
SOD (**A**,**E**), CAT (**B**,**F**), GR (**C**,**G**) and GPx (**D**,**H**) activity in the mitochondria-free cytosolic fraction from BG-Cereb (**A**–**D**) and C-Cortex (**E**–**H**) brain portions of CTRL, ROT and ROT + *SC*-Nanophyt groups. Error bars represent SEM for n = 5 independent experiments using five animals in each one. *, ** Significantly different from CTRL group, with *p* ≤ 0.05 and *p* < 0.01, respectively. ^##^, ^###^ Significantly different from ROT group, with *p* < 0.01 and *p* < 0.001, respectively.

**Figure 10 ijms-23-12699-f010:**
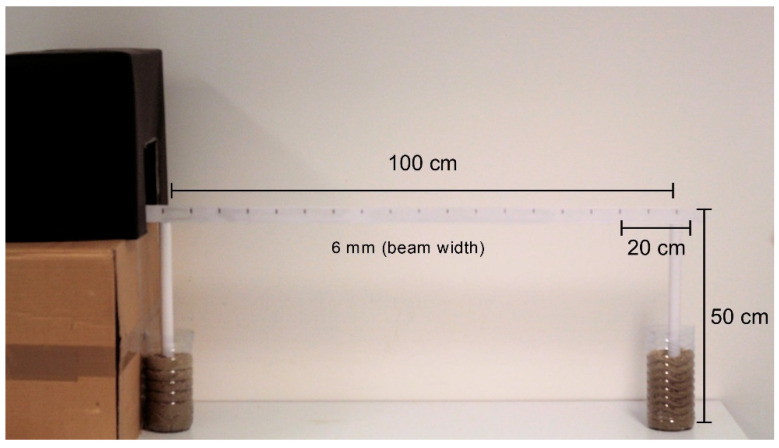
Image of the apparatus used for beam walking test.

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
