# Peer review of "Brain Effects of SC-Nanophytosomes on a Rotenone-Induced Rat Model of Parkinson’s Disease—A Proof of Concept for a Mitochondria-Targeted Therapy"

_ijms, 2022, doi:10.3390/ijms232012699_

Round 1
Reviewer 1 Report
Mendes D et al has demonstrated a promising mitochondrial targeted therapy using the SC-Nanophytosomes in the rotenone induced rat model of PD. Oral administration of SC-Nanophytosomes improved the motor symptoms of PD in the animal model. Authors have further demonstrated that SC-Nanophytosome, a nanosized vesicle, is quite stable in the rat GI tract after oral administration by drinking water. But authors did not talk about the brain penetrance/ crossing blood brain barrier of these nano vessels. Authors have investigated several mitochondrial parameters, like respiratory chain, electron transport chain, fatty acid profile in mitochondria-rich fraction of the animal brain. Treatment with SC-Nanophytosome improved over all symptoms of PD related motor dysfunction in the rotenone induced rat model. But Cannon JR et al., Neurobiol Dis. 2009 showed that rotenone treatment causes about 45% loss of tyrosine hydroxylase-positive neurons in the substantia nigra and a commensurate loss of striatal dopamine. This neuronal loss in the rat is irreversible. Therefore, how treatment with SC-Nanophytosome reverses the disease phenotypes was not clearly described. Overall experimental design and execution were well taken care off. But my suggestion to do some immunofluorescence study to show the neuronal loss as well the neuroprotection ability of SC-Nanophytosome in the substantia nigra of rat. After addressing these points, the manuscript can be considered for the publication.
Author Response
Answer to reviewers
We are grateful for reviewers’ careful manuscript appreciation, and we will try our best to address all the suggestions proposed. The modifications are highlighted in track changes in the new version of the manuscript.
Reviewer #1
Comment: Mendes D et al has demonstrated a promising mitochondrial targeted therapy using the SC-Nanophytosomes in the rotenone induced rat model of PD. Oral administration of SC-Nanophytosomes improved the motor symptoms of PD in the animal model. Authors have further demonstrated that SC-Nanophytosome, a nanosized vesicle, is quite stable in the rat GI tract after oral administration by drinking water. But authors did not talk about the brain penetrance/ crossing blood brain barrier of these nano vessels. Authors have investigated several mitochondrial parameters, like respiratory chain, electron transport chain, fatty acid profile in mitochondria-rich fraction of the animal brain. Treatment with SC-Nanophytosome improved over all symptoms of PD related motor dysfunction in the rotenone induced rat model. But Cannon JR et al., Neurobiol Dis. 2009 showed that rotenone treatment causes about 45% loss of tyrosine hydroxylase-positive neurons in the substantia nigra and a commensurate loss of striatal dopamine. This neuronal loss in the rat is irreversible. Therefore, how treatment with SC-Nanophytosome reverses the disease phenotypes was not clearly described. Overall experimental design and execution were well taken care off. But my suggestion to do some immunofluorescence study to show the neuronal loss as well the neuroprotection ability of SC-Nanophytosome in the substantia nigra of rat. After addressing these points, the manuscript can be considered for the publication.
Answer to comment:
We agree with the relevance of the reviewer's comment. First, we emphasize that the main goal of the present work is the assessment of the competence of SC-Nanophytosomes to support a mitochondria-targeted therapy to treat neurodegenerative diseases. For this, a rotenone-induced rat model of Parkinson's disease was used and, the experimental plan was designed to assess mainly effects on functionality of the mitochondrial respiratory chain and mitochondrial lipidome. Thus, it was not preserved brain samples to detect putative effects on tyrosine hydroxylase-positive neurons by using immunohistochemical techniques. However, to address the reviewer's issue the activity of the tyrosine hydroxylase, a rate-limiting enzyme in dopamine biosynthesis, was assessed in mitochondria-free cytosolic fraction in both brain portions. The obtained results were included in Figure 3C,D and discussed in Results and Discussion section (pages 6 and 7, lines 243-260) while the experimental procedure was included in the Material and Methods section (page 18, lines 616-629).
Regarding the “…the brain penetrance/ crossing blood brain barrier of these nano vessels” issue, the positive functional outcomes trigger by the oral administration of SC-Nanophytosomes on brain mitochondria of rotenone-induced PD animal model support the idea that both building blocks of the SC-Nanophytosomes, i.e., elderberry anthocyanins and algae lipids, reach the mitochondria of brain cells, as discussed in the first version of the manuscript (see in current version, page 9, lines 352-357 for anthocyanins; and page 10, lines 372-375 for algae lipids).
Considering the available results, we can suggest that SC-Nanophytosomes is an effective tool to provide elderberry anthocyanins and algae lipids for the brain cells, without any consideration about their competence to cross the several physiological barriers, including the blood brain barrier (this idea is included in the conclusion of the new version of the manuscript, see page 20, lines 718-724).
Recognizing the relevance of this subject, we can anticipate that a new investigation has already started, aiming: i) to unveil the cellular mechanisms used by SC-nanophytosomes to cross the blood brain barrier, using a cellular model of BBB; ii) to characterize the safety profile and the blood and brain pharmacokinetic parameters of SC-nanophytosomes oral administration as function of the dose, using to healthy animal and considering different time points after oral administration. Thus, only after the completion of this project, scheduled for July 2023, it will be possible to address all aspects underlying the reviewer's comment.
Reviewer 2 Report
Mendes et al,
The authors evaluate the effects of nanophytosomes as a drug delivery system on a rotenone-induced model of Parkinson’s disease (PD). They deliver this system to mice using drinking water, underlining the usefulness in potential treatment scenarios. Moreover, they show that such a drug delivery of elderberry anthocyanins improves motor function, decreases α-synuclein and antioxidative enzyme content in tissue and improves the fatty acid profiles of cells. I found the study to be an acceptable proof of principle of drug delivery in mice. The authors show that their experiments work on the particular systems tested and I am supportive of publication. I have only a few points to ask for explanation.
Major points:
1. The study is based on a rotenone-induced model of PD. Could the authors comment on what they would expect from other available models of PD? For example, Could a few lines be added to explain that the effects that they see will apply to any model, why or possibly why not, and that this method could be eventually helpful for actual patients as well?
2. What are the reasons for choosing algal lipids? Why not use more vertebrate/human polar lipids for this delivery?
3. Could the authors comment on the state of the animals upon nanophytosome treatment? Meaning, was there any adverse immune reaction to this treatment, like inflammation, general health problems in mice, etc.? If not, this may enhance their claim of this platform being a general drug delivery method.
Author Response
Answer to reviewers
We are grateful for reviewers’ careful manuscript appreciation, and we will try our best to address all the suggestions proposed. The modifications are highlighted in track changes in the new version of the manuscript.
Reviewer #2
General comment: The authors evaluate the effects of nanophytosomes as a drug delivery system on a rotenone-induced model of Parkinson’s disease (PD). They deliver this system to mice using drinking water, underlining the usefulness in potential treatment scenarios. Moreover, they show that such a drug delivery of elderberry anthocyanins improves motor function, decreases α-synuclein and antioxidative enzyme content in tissue and improves the fatty acid profiles of cells. I found the study to be an acceptable proof of principle of drug delivery in mice. The authors show that their experiments work on the particular systems tested and I am supportive of publication. I have only a few points to ask for explanation.
Major points:
- The study is based on a rotenone-induced model of PD. Could the authors comment on what they would expect from other available models of PD? For example, could a few lines be added to explain that the effects that they see will apply to any model, why or possibly why not, and that this method could be eventually helpful for actual patients as well?
Answer to major point 1:
We agree with the relevance of the reviewer's comments. Thus, to address the major point 1, the Conclusions section of the manuscript was modified to include some arguments that support the idea that SC-Nanophytosomes are also able to promote benefits in other PD animal models and human patients (see page 20, lines 726-734).
- What are the reasons for choosing algal lipids? Why not use more vertebrate/human polar lipids for this delivery?
Answer to major point 2:
It is an interesting question. In the manuscript introduction, we present our rationalization for the selection of both building blocks of SC-Nanophytosomes, as follow: “Elderberry anthocyanins were selected by their ability to oxidize NADH in the aqueous phase and deliver the electrons to complex III of the mitochondrial redox chain, bypassing the abnormalities associated with impaired complex I activity working as a membrane electron carrier [17]. C. tomentosum membrane polar lipids, with high levels of anionic phospholipids (including 22.5% of phosphatidylglycerol, which is a precursor of cardiolipins) and omega-3 polyunsaturated fatty acids (PUFA) [16,18], were selected to modulate the mitochondrial lipidome.” Now, we add other important reason, discussed in our previous article, and confirmed by the presented results, namely: the high levels of anionic phospholipids with polyunsaturated fatty acids, specific from chloroplast-rich tissues of green algae and plants, are required to preserve into the bilayer of the nanovesicles the elderberry anthocyanins in their mitochondriotropic cationic form [16] (see page 2, lines 82-91).
In the present letter we can also state that in animals, the mitochondria also exhibit lipids with high levels of anionic phospholipids with polyunsaturated fatty acids, but they are dominated by cardiolipins and not by PG. Unlike PGs, CLs are immunoreactive, and if a cardiolipin-rich nanoplatform fuses with the outer membrane of mitochondrial, it will raise the levels CL in this membrane with subsequent activation of the pathways underlying to mitophagy and/or apoptosis [see Nat. Cell. Biol. 2013, 15, 1197-1205. https://doi.org/10.1038/ncb2837], so the animal sources do not serve the purposes of the present research.
- Could the authors comment on the state of the animals upon nanophytosome treatment? Meaning, was there any adverse immune reaction to this treatment, like inflammation, general health problems in mice, etc.? If not, this may enhance their claim of this platform being a general drug delivery method.
Answer to major point 3:
We understand the relevance of this reviewer's comment. In fact, heart, kidneys, liver, and skeletal muscle samples were collected from three groups and preserved to study the effects of SC-Nanophytosomes treatment at level of peripheral organs, with relevance in Parkinson disease. This study will be also important to reveal aspects related to the safety and/or toxicological profile of the SC-Nanophytosomes formulation. In this letter we can advance that the results obtained with skeletal muscle confirm the brain benefits of SC-Nanophytosomes (described in the present article) and do not reveal signs of toxicity, but the investigation remains in progress. The set of results obtained with the peripheral organs will be subject of a new publication.
Results and Discussion section of the present manuscript was changed to include the following sentences (see page 5, lines 199-204). “It is also important to highlight that, after three weeks of treatment, the animals' body weight of the ROT+SC-Nanophyt group, as well as the weight of their peripheral organs (e.g., heart, kidneys, and liver) did not show significant differences compared with the CTRL group (Figure 2S and Table 1S of supplementary material). Additionally, these animals no longer display visual signs of postural instability and bradykinesia or of any other health problems (Figure 1S, supplementary material).”
Round 2
Reviewer 1 Report
This manuscript can be consider for the publication.